# FLASH Radiotherapy: Current Knowledge and Future Insights Using Proton-Beam Therapy

**DOI:** 10.3390/ijms21186492

**Published:** 2020-09-05

**Authors:** Jonathan R. Hughes, Jason L. Parsons

**Affiliations:** 1Cancer Research Centre, Department of Molecular and Clinical Cancer Medicine, University of Liverpool, 200 London Road, Liverpool L3 9TA, UK; jonathan.hughes@liverpool.ac.uk; 2Clatterbridge Cancer Centre NHS Foundation Trust, Clatterbridge Road, Bebington CH63 4JY, UK

**Keywords:** FLASH, ionizing radiation, proton beam therapy, radiotherapy, radiobiology

## Abstract

FLASH radiotherapy is the delivery of ultra-high dose rate radiation several orders of magnitude higher than what is currently used in conventional clinical radiotherapy, and has the potential to revolutionize the future of cancer treatment. FLASH radiotherapy induces a phenomenon known as the FLASH effect, whereby the ultra-high dose rate radiation reduces the normal tissue toxicities commonly associated with conventional radiotherapy, while still maintaining local tumor control. The underlying mechanism(s) responsible for the FLASH effect are yet to be fully elucidated, but a prominent role for oxygen tension and reactive oxygen species production is the most current valid hypothesis. The FLASH effect has been confirmed in many studies in recent years, both *in vitro* and *in vivo*, with even the first patient with T-cell cutaneous lymphoma being treated using FLASH radiotherapy. However, most of the studies into FLASH radiotherapy have used electron beams that have low tissue penetration, which presents a limitation for translation into clinical practice. A promising alternate FLASH delivery method is via proton beam therapy, as the dose can be deposited deeper within the tissue. However, studies into FLASH protons are currently sparse. This review will summarize FLASH radiotherapy research conducted to date and the current theories explaining the FLASH effect, with an emphasis on the future potential for FLASH proton beam therapy.

## 1. Introduction

Radiotherapy using X-rays (photons) is a main treatment strategy employed to combat human tumors, with ~50% of all cancer patients receiving radiotherapy. However, the major drawback of radiotherapy treatment is that in order to deliver a lethal dose to cancerous cells, short- and long-term adverse side-effects are evident due to the irradiation of the surrounding normal healthy tissues that can severely impact the health and quality of life of the cancer patient [1,2,3]. This occurs because radiotherapy uses an external radiation beam where the dose decreases exponentially but which can deposit energy within a certain depth of the patient tissue [4]. Therefore, in the cases of deep-seated tumors, the healthy normal tissue in front of the tumor receives a large dose of ionizing radiation relative to the tumor. Furthermore, it is possible that healthy normal tissue located behind the tumor can receive an exit dose of radiation if the beam passes through the tumor. This can present significant challenges to sensitive tissues and organs at risk, such as the brain and spinal cord. Advancements in modern radiotherapy deliverance and imaging techniques such as image-guided radiotherapy, intensity-modulated radiotherapy, and volumetric modulated arc therapy, along with targeted combinatorial drug therapies and immunotherapy, have increased the therapeutic index of radiotherapy [5,6,7,8,9,10]. Furthermore, the increased use of proton beam therapy (PBT) which displays a lower entrance dose compared to conventional radiotherapy and where the majority of the radiation dose can be specifically targeted at the tumor, can also limit the unnecessary irradiation of surrounding normal tissues leading to reduced adverse side-effects [11]. Despite this, many tumors remain intrinsically radioresistant and therefore further discovery and research into novel treatment strategies is critical to maximize the tumor-killing effect of radiotherapy, while simultaneously minimizing the toxic impact to surrounding normal tissues.

Excitingly, a recent series of research studies examining “FLASH” irradiation, a term first coined in 2014 by Favaudon et al. and the Vozenin group in Lausanne, has demonstrated that it possesses a normal tissue sparing capability while maintaining tumor cytotoxicity when compared to conventional radiotherapy in several *in vivo* models [12,13,14,15]. FLASH irradiation is the deliverance of dose at ultra-high dose rates (>40 Gy/s) that are several orders of magnitude higher than conventional dose rates (~5 Gy/min) that are used clinically. The normal tissue sparing phenotype is consequently a phenomenon called the “FLASH effect”. Despite the spike in FLASH interest in recent years, the advantages of using ultra-high dose rate radiotherapy and the FLASH effect was originally reported as far back as 1960–1970, although further investigations were largely halted due to logistical difficulties translating the findings into clinical practice [16,17]. However, with the improvements in modern-day technology and a greater understanding of radiobiology, FLASH is demonstrating potential as a key tool in the future of clinical radiotherapy. Before this can happen though, it is critical that the underlying biological mechanisms and optimal beam delivery parameters are realized, as these currently remain largely uncovered.

## 2. The FLASH Effect

### 2.1. Normal Tissue Sparing

The FLASH effect is defined as the decrease in radiation-induced normal tissue toxicities with dose delivery at ultra-high dose rates (FLASH), compared to conventional dose rates used clinically. The FLASH effect has now been observed in several *in vitro* cellular models, and multiple *in vivo* animal models (summarized in Table 1). The earliest report of the FLASH effect was described in 1966 where it was discovered that mice irradiated at ultra-high dose rates had a greater survival than those irradiated at conventional dose rates [17]. A renewed interest in FLASH peaked more recently in 2014 in which C57BL/6J mice were comparatively treated with 17 Gy of either FLASH dose rates (60 Gy/s, 4.5 MeV electrons) or conventional dose rates (0.03 Gy/s, γ-rays or 4.5 MeV electrons) and the presence of lung fibrogenesis was investigated [12]. Here, mice were observed for up to 36 weeks following bilateral thorax irradiation and for the conventional dose-rate-treated mice, pulmonary fibrosis developed as early as 8 weeks and progressively worsened up to 36 weeks. By contrast, mice treated with 17 Gy FLASH dose rates were relatively free of pulmonary fibrosis, and doses of 30 Gy FLASH irradiation were required to induce significant fibrosis which was seen with 17 Gy at conventional dose rates [12]. In addition to the lungs, the FLASH effect has also been investigated and confirmed in several other organs using mouse models, including brain, skin and gut [13,18,19,20,21].

Recent data investigating the neurocognitive development of juvenile (3-week old) mice showed a radioprotective FLASH effect following 8 Gy whole brain irradiation with ultra-high FLASH dose rates (4.4 × 10^6^ Gy/s, 6 MeV electrons) when compared to conventional dose rates (0.077 Gy/s, 6 MeV electrons) [22]. Mice were subjected to several neurocognitive tests following irradiation and in all cases the performance of the FLASH-irradiated animals was indistinguishable from the control group, whereas conventional irradiation caused a significant detriment. It was suggested that the neurocognitive benefits of FLASH irradiation was due to a preservation of the neurogenic niche and neurogenesis in the FLASH treated mice, with conventional dose-rate-irradiated mice showing considerably lower levels of immature and mature neurons four months post-irradiation. Furthermore, the long-term benefits of FLASH on pituitary function was also investigated and it was found that 8 Gy conventional dose-rate-irradiated mice had a two-fold reduction in levels of plasma growth hormone levels one-week post-treatment compared to the non-irradiated controls, whereas no significant decrease was observed in the FLASH-irradiated animals [22].

Aside from mice, the FLASH effect has also been confirmed in mini-pigs and cats, higher animal models that are more similar to humans [14]. Pig skin irradiated at the same time with either FLASH (300 Gy/s) or conventional dose-rate (0.083 Gy/s) radiation were used to comparatively investigate the difference in cutaneous lesions formed. The experiment was performed by irradiating multiple 26 mm diameter circular patches on the skin of a single mini-pig with single doses ranging from 22–34 Gy. Over the course of 48 weeks, the clinical pathologies following skin irradiation including, depilation/destruction of hair follicles, fibronecrosis, epithelial ulceration, and inflammation, were observed with the conventional dose-rate treatments. However, the results observed following FLASH remained comparable to that of non-irradiated skin, showing only minor depilation and pigmentation and therefore starkly different to conventional dose rates. Furthermore, it was suggested that the dose modifying factor was >1.36 for FLASH compared to conventional dose rates using the absence of late stage necrosis at 9 months as an endpoint, whereby similar results were obtained for 34 Gy FLASH and 25 Gy at conventional dose rates [28].

### 2.2. Tumor Control

An important attribute of FLASH that has been reported in only a limited number of studies, is the ability to generate a similar anti-tumor response as the equivalent conventional dose-rate radiation (summarized in Table 2). This potentially means that larger doses could be administered to radioresistant tumors using FLASH radiotherapy due to the increased therapeutic index. For example, in breast (HBCx-12A) and head and neck cancer (Hep-2) xenograft models, FLASH was found to be as efficient at controlling tumor growth as conventional radiotherapy [12]. In parallel, an orthotopic lung tumor model using luciferase-positive TC-1 cells injected into C57BL/6J mice, revealed no observed difference in anti-tumor efficiency when mice were exposed to either FLASH or conventional radiotherapy. In a subsequent dose-escalation experiment, it was observed that at 8–9 weeks post-irradiation, only 20% of 15 Gy conventional dose-rate radiotherapy treated mice were tumor free, whereas 70% of the 28 Gy FLASH treated mice were free of tumors. Furthermore, the conventional radiotherapy treated mice displayed inflammatory and fibrotic remodeling, whereas the FLASH treated mice did not [12].

In the same study described above investigating mini-pig skin, the impact of FLASH electrons on six cats with advanced squamous cell carcinoma of the nasal planum were examined, although conventional dose rates were not comparatively used [14]. Each of the six cats were given a single individual dose ranging from 25–41 Gy, and it was reported that the cats responded very well, with only mild dermatitis/mucositis observed and no late stage toxicities. In terms of the tumor, 5 of the 6 cats achieved complete remission by 16 months, with one cat experiencing a local recurrence at 21 months. Although the images presented within this study are striking, and the overall results are promising, it is limited by the lack of a control using conventional dose-rate radiotherapy, so comparisons analyzing the anti-tumor control of FLASH versus conventional dose rates was not reported.

Interestingly, the first patient to be treated with FLASH radiotherapy has been performed at the Lausanne University Hospital [15]. The patient was a 75-year old male that presented with CD30+ T-cell cutaneous lymphoma that he was diagnosed with in 1999, and over the course of a ten-year period (2008–2018), the patient had received localized radiotherapy that generally controlled the lymphoma but he experienced severed acute toxicity to the surrounding skin. For the FLASH treatment, the 3.5 cm tumor was treated with a 15 Gy total dose delivered over 10 × 1 μs pulses (≥10^6^ Gy/s, 1.5 Gy per pulse) with a total treatment time of 90 ms. Initial tumor shrinkage began after 10 days, and a complete tumor response was achieved at 36 days that was preserved for 5 months. In terms of toxicity to the surrounding skin, redness, mild epithelitis and edema (grade 1) was observed at 10–12 days post-irradiation with a maximal reaction at 3 weeks, which was deemed mild and healed much quicker compared to the patients previous localized radiotherapy treatments. Although this report shows promising data demonstrating the feasibility of FLASH radiotherapy in the clinic, as well as the observed FLASH effect and positive patient outcome, FLASH is not yet ready to be fully translated for cancer patient treatment. Larger patient trials comparing conventional dose-rate radiotherapy with FLASH still need to be performed, along with investigations into the appropriate radiation sources and equipment that can treat tumors other than superficial skin tumors.

## 3. Mechanisms Contributing to the FLASH Effect

### 3.1. Oxygen Depletion

The exact biochemical mechanisms that result in the FLASH effect are yet to be fully elucidated, although the current theory gaining the most ground implicates oxygen as a critical molecule in the biological response to FLASH irradiation [30]. Generally in response to ionizing radiation, indirect DNA damage occurs through the radiolysis of water and subsequent generation of reactive oxygen species (ROS), such as hydroxyl radicals, that attack the DNA [31]. It is estimated that for low linear energy transfer (LET) radiation, such as photons and electrons, 60–70% of the DNA damage induced is through generation of ROS whereas 30–40% is via direct interaction of the radiation with DNA [32,33]. The oxygen fixation hypothesis suggests that if this indirect DNA damage is a result of reaction with a free radical (e.g., hydroxyl radical), the damage is fixed due to the presence of molecular oxygen through the formation of a more damaging peroxyl radical [34]. Indeed, this is a major contributor as to why hypoxic tumors are more radioresistant than well-oxygenated tumors that display an oxygen enhancement ratio of ~2–3 [35,36]. In terms of FLASH, the oxygen depletion hypothesis suggests that the ultra-high dose rate modulates the immediate radiochemical events that occur in the irradiated tissue [37]. In this short exposure time frame, local oxygen is depleted faster than reoxygenation can occur, leading to a transient state of radiation-induced hypoxia, and therefore radioresistance and protection of the normal tissues to the FLASH irradiation [38].

The relationship between increasing dose rates and the oxygen depletion hypothesis was realized in early bacterial studies, whereby irradiation at ultra-high doses produced a survival curve indicative of those irradiated in an anaerobic (hypoxic) environment [39,40,41,42]. A subsequent study also suggested oxygen depletion as the reason why there was resistance in the tails of mice irradiated at high dose rates to epithelial necrosis [43]. However, *in vitro* evidence using mammalian cell lines to observe the FLASH effect have been lacking, with mixed reports as to whether this phenomenon was observed or not [16,44,45,46,47,48]. This can be explained, in part, due to FLASH studies using cells cultured in atmospheric oxygen concentrations (~20%), whereas the normal tissue sparring observed *in vivo* is generally at physiological oxygen tensions from 3–7%. This means that the FLASH doses used in these *in vitro* studies were not sufficient enough to significantly reduce the oxygen tension [28,49,50]. In support of this, a recent study using prostate cancer cells irradiated at 600 Gy/s (10 MeV electrons) showed significant survival versus conventional dose-rate irradiation (14 Gy/min) at oxygen concentrations of 1.6%, 2.7% and 4.4%, but no significant difference was seen at higher oxygen concentrations of 8.3% and 20% [45]. The oxygen depletion hypothesis does raise an important issue whether FLASH can be translated clinically. This is because for tumors that contain a heterogonous population of cells at different oxygen concentrations, the FLASH effect may be therapeutically detrimental by actually increasing tumor radioresistance. Therefore, the role of oxygen tension and the impact on FLASH radiotherapy must be explored in more detail experimentally.

### 3.2. ROS

Other oxygen-related products, including ROS and free radicals, have been theorized to have an altered biochemistry between normal tissue and tumors, thus contributing to the FLASH effect. In an experiment conducted using zebrafish embryos following conventional dose rate (0.1 Gy/s) or FLASH (1 pulse of 1.8 × 10^−6^ s) electron irradiation, it was concluded that FLASH led to less of an effect on zebrafish morphology 5 days post-fertilization due to a lower production in ROS [23]. However, if the zebrafish were incubated with the ROS scavengers, amifostine, or *N*-acetyl-cysteine 1 h prior to irradiation with conventional or FLASH radiotherapy, body length measurements 5 days post-fertilization revealed no significant difference between the FLASH or conventional radiotherapy treated zebrafish. Overall, this study demonstrated that FLASH offers radioresistance in normal tissue due to reduced ROS levels. Differences in redox chemistry and free radical production have recently been used to explain the contrasting biological effects between normal and cancer tissue following FLASH [33]. It has been hypothesized that due to normal cells having lower pro-oxidant burdens during normal redox metabolism and an increased ability to sequester labile iron compared to cancerous cells, normal cells can more effectively reduce the levels of free radicals and hydroperoxides generated from peroxidation chain reaction and Fenton type chain reactions following FLASH, therefore increasing the oxidative burden in cancer cells [33].

### 3.3. Immune Response

The inflammatory and immune responses have also been suggested as underlying mechanisms that contribute to the FLASH effect. Transforming growth factor beta (TGF-β), an important pro-inflammatory cytokine, has particularly been implicated to alter the effects of FLASH compared to conventional dose-rate radiotherapy. In an *in vitro* study using proton irradiation, the induction of TGF-β levels in human lung fibroblasts were significantly reduced following 20 Gy FLASH (1000 Gy/s) versus conventional dose rates (0.2 Gy/s) [47]. For the FLASH dose rate, a ~1.8-fold induction in TGF-β levels was observed 24 h post-irradiation, while a ~6.5-fold increase was observed following conventional dose rates, suggesting that FLASH may have the potential to reduce radiation-induced chronic inflammation. A reduction in TGF-β signaling was also previously reported for FLASH-irradiated mice versus conventional dose rates [12]. In support of a shifting balance from a pro-inflammatory towards an anti-inflammatory phenotype, a study investigating whole brain irradiation of C57BL/6J mice showed a reduction in hippocampal pro-inflammatory cytokine levels following FLASH compared to conventional dose-rate irradiation [20]. It was reported that at 10 weeks post-irradiation, there was a statistically significant increase in five out of ten cytokines tested following conventional dose rates, whereas FLASH generated an increase in only three cytokines.

In general, the role of TGF-β and its associated signaling pathway is known to be involved in the anti-tumor immune response following conventional radiotherapy, although the precise effects are still debated [51]. One study has reported that TGF-β is key to the radioresistance of tumor infiltrating T-cells [52], while others suggest TGF-β signaling suppresses the immune system and promotes cancer progression, pushing the need for the use of TGF-β pathway inhibitors [53]. Consequently, the alterations observed in TGF-β signaling and immune system activation following FLASH irradiation need to be carefully considered for clinical translation of FLASH, especially when radiotherapy is combined with immunotherapy. It has also been suggested that FLASH may offer an improved immune response due to the fast exposure time leading to less irradiation of circulating immune cells, although this effect may be reduced for fractionated FLASH radiotherapy [54]. Finally, it has been reported that proton irradiation of mice at FLASH dose rates showed an increased T-lymphocyte recruitment into the tumor microenvironment compared to conventional dose rates, further supporting the notation that changes in the immune response may contribute to the FLASH effect [29].

## 4. The Potential for FLASH Proton-Beam Therapy

Although in the simplest terms FLASH is the use of radiation dose rates multiple orders of magnitude higher than conventional dose rates, several other factors need to be taken into consideration to elicit the FLASH effect. Along with dose rate, these factors include total dose delivered, pulse rate/duration/width/number and total delivery time. Another important parameter is the irradiation source, with many of the current FLASH investigations using electron linear accelerators [12,14,15,55,56]. However, these experimental electron beams are currently limited to treatment of superficial cancers and intraoperative radiation therapy due to the low tissue penetration and limited field size of these beams (~4–20 MeV) [8]. On the other hand, clinical PBT offers a much greater tissue penetration and allow the irradiation of more deep-seated tumors. The significant advantage of PBT over conventional photon radiotherapy is that the majority of the beam energy is deposited in a narrow range called the Bragg peak following a low entrance dose (Figure 1), allowing the precise targeting of the tumor volume while sparing normal healthy tissue and organs at risk [11]. As a result, there has been an increase in the clinical use of PBT with ~150,000 cancer patients being treated to date. However, there are still significant biological uncertainties following proton irradiation largely due to the increases in LET at and around the Bragg peak, leading to changes in the DNA damage spectrum and increases in the radiobiological effectiveness [11]. Radiobiological research has also been impeded by the lack of accessible proton facilities for *in vitro* and *in vivo* experimentation. Despite this, the promise of proton FLASH has been invested in by multiple companies such as Varian, IBA and Mevion who are funding both the development of FLASH PBT machinery and research [57].

### Studies Investigating FLASH Protons

Current research on FLASH protons conducted *in vitro* and *in vivo* has revealed mixed information as to whether the FLASH effect was induced or not. In general, *in vitro* studies investigating FLASH protons have produced a lack of positive results observing the FLASH effect, particularly in terms of acute endpoints, such as clonogenic survival, γH2AX foci formation and cell cycle arrest. Data from such studies has been recently reviewed, and only one of ten studies demonstrated evidence of a FLASH effect [58]. Interestingly, though, all these studies were performed at aerobic oxygen levels (21%) and it is likely that this is the major reason for the absence of the FLASH effect. It is therefore clear that in order to investigate the FLASH effect *in vitro* with protons, experiments at varying oxygen tensions need to be performed, similar to those performed with electrons [45,50]. Regarding the one study reporting positive results *in vitro*, this was conducted using normal human lung fibroblasts (IMR90) and comparing conventional dose rate (0.05 Gy/s) and FLASH (100 or 1000 Gy/s) proton irradiation (4.5 MeV). It was observed that the increasing dose rate reduced the number of prematurely senescent cells (measured using β-galactosidase positive cells) and also reduced the induction of TGF-β expression, suggesting a long-term role of the FLASH effect particularly on chronic inflammation [47]. However, the difference in proton dose rate was concluded to have little effect on acute biological outcomes, including clonogenic survival and γH2AX foci formation. In fact, there was an indication of decreased clonogenic survival at both the FLASH dose rates compared to the conventional dose rate, albeit the data was not statistically significant. Interestingly, significantly less γH2AX foci formation was observed only following 20 Gy of 1000 Gy/s FLASH compared to 100 Gy/s FLASH and the conventional dose rates, potentially suggesting a role for reduced yields of double strand breaks (DSBs) and an altered DNA repair capacity following FLASH protons but only at very high doses [47]. It is worth noting that changes in the DNA damage response following FLASH in general are surprisingly understudied considering this is a major factor in radiobiology. Indeed, impacts on endpoints such as cell cycle progression, chromosomal aberrations, ROS levels, as well as DNA damage signaling and DNA damage foci relating to DSB formation following FLASH could be key to further understanding the underlying mechanisms that cause the FLASH effect. Another important consideration, which has not been investigated up to now, is whether the FLASH effect is still observed with increasing LET at and around the Bragg peak, and whether the profile of DSBs and complex DNA damage induced is altered [11]. We have recently demonstrated that complex DNA damage, containing multiple DNA lesions including oxidative DNA base damage and DNA single strand breaks within close proximity (1–2 helical turns of the DNA), along with DSBs, are a critical factor in radiation-induced cell killing and which triggers a specific cellular DNA damage response [59,60]. Therefore, it is important to determine the DNA damage profile with FLASH protons at higher LET.

Regarding *in vivo* studies, one report has investigated morphological changes in zebrafish embryos with either conventional dose rates (5 Gy/min) or FLASH (100 Gy/s) [61] but could not replicate FLASH sparing with protons that was previously observed with electrons [23]. A possible reason for not observing the FLASH effect was suggested due to the proton-beam pulse characteristics that delivered a lower maximum dose rate per pulse. Micro-pulse dose rates delivered by the cyclotron were ~10^3^ Gy/s, whereas electron macro-pulse dose rates have been reported as ~10^7^ Gy/s. Furthermore, the zebrafish embryos in this study were irradiated at a later developmental stage post-fertilization compared to the previous electron-focused study (~24 hpf vs. 4 hpf), potentially making the zebrafish less sensitive to FLASH PBT irradiation and contributing to the lack of an observed FLASH effect. However in general, more recent *in vivo* studies investigating FLASH PBT have yielded much more positive findings by observing the FLASH effect, and associated tumor control compared to using conventional dose rates (summarized in Table 3). In an innovative study, a clinical 230 MeV proton accelerator using double-scattered protons under CT guidance was designed to deliver FLASH dose rates of 60–100 Gy/s and conventional dose rates of 0.5–1 Gy/s [27]. Here, 8–10-week-old C57BL/6J mice were subjected to whole abdominal irradiation with 15 Gy of either FLASH (78 Gy/s) or conventional dose-rate (0.9 Gy/s) protons and intestinal segments were harvested 3.5 days post-irradiation. It was found that FLASH significantly reduced the loss of proliferating intestinal crypt cells versus conventional dose-rate radiotherapy. In addition, mice irradiated with 18 Gy protons focused on the intestines and harvested 8 weeks post-irradiation, revealed that conventional dose-rate-irradiated mice had considerably increased fibrosis compared to the FLASH-irradiated mice. The degree of fibrosis following FLASH PBT treatment was actually comparable to that of unirradiated mice. Finally, MH641905 pancreatic tumor cells injected into the mice to generate flank tumors were irradiated with 12 or 18 Gy FLASH alongside conventional dose-rate irradiation, and no significant difference between tumor growth delay was observed between the treatments. Therefore, these results demonstrate normal tissue sparring along with effective tumor control with FLASH PBT, at least for gastrointestinal tumors [27].

Several other mouse model studies have shown the benefits of FLASH PBT. In the first, whole thorax irradiation (15–20 Gy) was delivered to C57BL/6J mice using FLASH (40 Gy/s) or conventional (1 Gy/s) dose rates, and responses analyzed at 8–34 weeks post-irradiation [25]. FLASH tissue sparring was observed through a 30% reduction in lung fibrosis, reduced skin dermatitis, and improved overall survival in the FLASH PBT treated mice. In addition, genome-wide microarray analysis was performed in order to uncover the underlying mechanisms involved in the FLASH effect, with this demonstrating that DNA repair, inflammation, and the immune response as the major pathways differentially regulated between the two PBT dose rates. In a related study, the whole thorax region of C57BL/6J mice were irradiated with 15–20 Gy using a clinical pencil-beam scanning PBT system with either FLASH (40 Gy/s) or conventional dose-rate protons (0.5 Gy/s), and the mice analyzed at 8–36 weeks post-irradiation. Surprisingly, gender-specific differences were observed with only the female mice cohort showing improved outcomes following FLASH [24]. Nevertheless, these mice displayed better breathing function, reduced dermatitis, altered lung pathology, and greater overall survival with FLASH radiotherapy compared to using conventional dose rates. Finally, a study injected Lewis lung carcinoma (LLC) cells into the left lung of C57BL/6J mice and the whole lungs were irradiated with an 18 Gy dose of protons delivered using a clinical pencil-beam scanning PBT system at either FLASH or conventional dose rates. Tumor sizes were measured 7 days post-irradiation by imaging, and then at 10 days post-irradiation when the mice were sacrificed. Remarkably, it was observed that the lung tumors in the FLASH PBT irradiated mice were significantly smaller in comparison to conventional dose rates, suggesting in this case that FLASH protons have an increased tumor control capability compared to protons used at conventional dose rates [29]. Nevertheless, additional and more comprehensive *in vivo* studies examining FLASH PBT using the appropriate tumor models need to be conducted.

## 5. Conclusions

FLASH radiotherapy is an exciting new treatment strategy that has the potential to change the future of clinical cancer treatment. The use of ultra-high dose rates several orders of magnitude higher than conventional dose rates generates a phenomenon known as the “FLASH effect”, through which sparing of normal healthy tissue is observed, while maintaining equivalent tumor control properties compared to conventional dose-rate radiotherapy. Current radiotherapy regimes are limited by the tolerance of surrounding normal tissues to radiation-induced toxicities, meaning that some radioresistant tumors may not receive the required dose of radiation for the treatment to be effective. However, FLASH radiotherapy has the potential to overcome this and allow an increased radiation dose delivered to tumors while keeping the toxicity to surrounding healthy tissues low. Remarkably, the first patient with CD30+ T-cell cutaneous lymphoma has recently been treated using FLASH radiotherapy. It is clear that oxygen plays a key role in the underlying biological mechanism resulting in the FLASH effect. In fact, multiple studies have found that the ultra-high dose rate radiation is able to deplete local oxygen and induce a short-lived protective hypoxic environment within the normal healthy tissues that increases radioresistance. Furthermore, theories have suggested changes in ROS and redox chemistry between normal and tumor cells following FLASH dose rates. Although the oxygen depletion hypothesis is the most popular current explanation for the FLASH effect, other phenomena may play an important role, including the immune response and tumor microenvironment that require further examination. Despite this, an area that has been surprisingly understudied is whether there are any differences in the DNA damage profile (e.g., actual numbers and ratios of DNA base damage, DNA single strand breaks and DSBs) and the subsequent DNA damage response following FLASH irradiation (photons/electrons and PBT), in comparison to conventional dose rates. Therefore, future studies should focus on quantifying the levels and persistence of particularly DSBs and complex DNA damage (measured directly or using DNA damage foci) that are the key drivers contributing to the therapeutic effect of radiotherapy, in the appropriate 3D *in vitro* (spheroids/organoids) and/or *in vivo* models. Additionally, the DNA repair pathways responsive to FLASH-induced DNA damage, particularly non-homologous end-joining or homologous recombination involved in DSB repair, should be monitored. It is important to consider both the FLASH effect on sparing of normal cells/tissues, but also its impact in tumor cell killing, as well as appreciation of the oxygen levels at which the experiments are conducted. Nevertheless, it is likely that a myriad of biological changes are observed following FLASH irradiation.

Although the FLASH effect in theory appears revolutionary, translation into the clinic is still difficult at this early stage. This is because several factors contribute to the FLASH effect, including total dose, pulse rate, pulse duration, pulse width, pulse number, and total delivery time. Based on *in vitro* and *in vivo* reported data, doses upwards of tens of Gy are required to induce FLASH radioprotection, which can be too high to treat a significant number of patients clinically. Furthermore, questions arise as to whether a fractionation regime for FLASH to deliver higher doses will be able to induce a FLASH effect. Another question that needs to be answered is which source of radiation is best to deliver FLASH radiotherapy. Much of the current data has used electron sources, however this is currently limited to treatment of superficial cancers or intraoperative radiation therapy. PBT may offer the best solution to be able to treat some deep-seated tumors, and there are several high-energy clinical PBT facilities already in place that can be modified to generate FLASH dose rates [63]. Furthermore, several innovative set-ups are already being tested using modified clinically available PBT beams [27,64]. However, implementation of FLASH PBT still has its technical limitations. To deliver protons to a large tumor volume, the proton beam must be scattered which may cause particle loss and decrease the total dose delivered. Pencil-beam scanning enables the delivery of ultra-high dose rates per individual spot, however the time taken to perform this is extended, therefore reducing the total dose rate which may not be enough to induce the FLASH effect [37]. Furthermore, research output using protons has produced largely mixed results, and it is also still unknown how the increasing LET at and around the Bragg peak will impact on the FLASH effect. Therefore, significantly more research into FLASH PBT is required, particularly investigations at physiological oxygen concentrations, for this to be potentially translated into the clinic for the benefit of cancer patients.

## Figures and Tables

**Figure 1 ijms-21-06492-f001:**
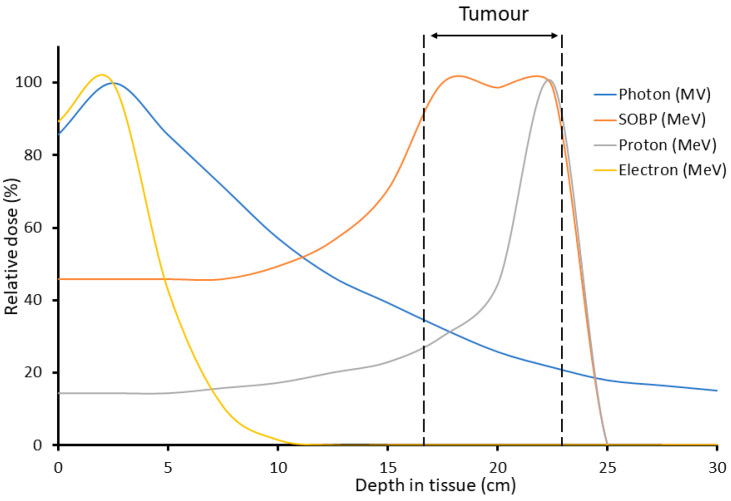
Comparison of depth–dose distribution of photons, protons, and electrons relative to a target tumor. A spread-out Bragg peak (SOBP) from several modulated proton beams is also shown, but which demonstrates the precise targeting of the tumor using PBT.

**Table 1 ijms-21-06492-t001:** Evidence of normal tissue sparing from FLASH irradiation.

Model (Site of Irradiation)	Assay/Endpoint	Dose (Gy)	Dose Rate (Gy/s)	Radiation Source	Reference
Mice (WBI) ^1^	Memory tests, neurogenesis	10	>100	Electron	[13]
Mice (WBI) ^1^	Neurocognitive tests, mature/immature neurons, growth hormone levels	8	4.4 × 10^6^	Electron	[22]
Mice (WBI) ^1^	Neurocognitive tests, dendritic spine density, microglial activation, inflammation	30	200/300	Electron	[20]
Mice (WBI) ^1^	Neurocognitive tests, neuroinflammation, neuronal morphology	10	>100	Electron	[23]
Mice (WBI) ^1^	Neurocognitive tests, hippocampal cell division, astrogliosis	10	37	X-ray	[21]
Mice (thorax)	Survival, dermatitis, breathing function, lung pathology	15/17.5/20	40	Proton	[24]
Mice (thorax)	Lung fibrosis, skin dermatitis, survival	15/17.5/20	40	Proton	[25]
Mice (thorax)	Lung fibrosis, TGF-β signaling, apoptosis	17	40–60	Electron	[12]
Mice (thorax)	Cellular proliferation, pro-inflammatory gene expression, DNA damage (53BP1/γH2AX foci), senescence	17	40–60	Electron	[26]
Mice (abdomen)	Survival	10–22	70–210	Electron	[19]
Mice (abdomen)	Survival, stool production, crypt cell regeneration, apoptosis, DNA damage	12–16	216	Electron	[18]
Mice (abdomen)	Intestinal crypt cell proliferation	15 Gy	78	Proton	[27]
Mice (local intestinal)	Fibrosis	18 Gy	78	Proton	[27]
Mini-pig (skin)	Skin toxicity/injury	22–34	300	Electron	[14]
Zebrafish Embryo	Morphology	8	>100	Electron	[23]

^1^ WBI refers to whole brain irradiation.

**Table 2 ijms-21-06492-t002:** Evidence of tumor control from FLASH irradiation.

Model	Assay/Endpoint	Dose (Gy)	Dose Rate (Gy/s)	Radiation Source	Reference
Mice, HBCx-12A, and Hep-2 human xenografts (local)	Tumor growth	17–25	60	Electron	[12]
Mice, orthotopic engrafted lung carcinoma luciferase+ TC-1 cells (thorax)	Tumor growth	15–28	60	Electron	[12]
Mice, ID8 syngeneic ovarian cancer (thorax)	Tumor number/weight	14	216	Electron	[18]
Mice, orthotopic engrafted Lewis lung carcinoma (thorax)	Tumor size	18	40	Proton	[29]
Mice, pancreatic MH641905 flank tumor	Tumor growth	12/15	78	Proton	[27]
Cat, nasal planum SCC (local)	Tumor growth	25–41	130–390	Electron	[14]
Human, CD30+ T-cell cutaneous lymphoma	Tumor response	15	167	Electron	[15]

**Table 3 ijms-21-06492-t003:** Summary of outcomes in in vivo studies comparing FLASH and conventional dose-rate PBT.

Model	Dose (Gy)	FLASH Dose-Rate (Gy/s)	Outcome	Reference
Zebrafish embryo	0–43	100	No survival difference	[61]
Mice (thorax)	15/17.5/20	40	Normal tissue protection with FLASH	[24]
Mice (thorax)	15/17.5/20	40	Normal tissue protection with FLASH	[25]
Mice (abdomen)	15	78	Normal tissue protection with FLASH	[27]
Mice (local intestinal)	18	78	Normal tissue protection with FLASH	[27]
Mice, orthotopic engrafted Lewis lung carcinoma (thorax)	18	40	Improved tumor control with FLASH, increased T-lymphocyte tumor infiltration	[29]
Mice, pancreatic MH641905 flank tumor	12/15	78	No difference in tumor control	[27]
Mice, FaDu head, and neck squamous cell carcinoma transplantation	17.4	>10^9^	No difference in tumor control	[62]

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
