# Peer review of "FLASH Radiotherapy: Current Knowledge and Future Insights Using Proton-Beam Therapy"

_ijms, 2020, doi:10.3390/ijms21186492_

Round 1
Reviewer 1 Report
This is an excellent overview of the existing literature on the potential for FLASH treatments for cancer by two experts in this area. The authors discussion in vitro, in vivo and clinical studies which good understanding of the variables at play in these studies. The overview covers various radiation modalities including x ray, electron, and protons pointing out inherent limitations with each. The author point out the current best explanation depends on the oxygen depletion effect of the FLASH dose and further this is only effective at low oxygen tensions. It is especially significant that fibrosis is greatly diminished in some FLASH studies. They point out that other phenomenon such as the immune response and the tumour microenvironment may play a role and need further examination. Such an incisive review is needed at this time to aid in determining the next steps to be taken in this area.
Minor comments:
page 1 line 54 introduction The authors need to point out the does is exponentially decreasing.
page 10 line 380 Here the authors state that "doses in the range of tens of Gy are required to induce FLASH radioprotection, which are too high to treat patients clinically? They should revise this as total doses of 10 Gy or higher are common clinically.
Reviewer 2 Report
This review manuscript targets the importance of the upcoming radiation treatment technique of FLASH for protons or electrons. That means a delivery of high dose over a small period of time like sec. The FLASH shows great results so far when treating a tumor for tumor control and adverse effects which usually relate mainly to an enhanced immune response in a chronic pattern I. e. overactive immune system and oxidative stress.
The review is nice in a relative conclusive and concise way although the topic is not with solid data on the underlying mechanisms and overall effectiveness for human patients. The authors touch and discuss the hypothetical redox, ROS or immune response theories. Here critical articles connecting dna damage induction with immune system triggering and Danger signals should be discussed as in 2019 Nov 14;11(11):1789.doi: 10.3390/cancers11111789 and other works showing a great association between DNA damage induction and immune system through DDR.
In addition a better description of complex and clustered DNA damage induction should be provided. OCDLs or MDS should be mentioned and explained.
But why really one expects a different pattern of damage with FLASH.? The authors have to discuss previous studies comparing low and high dose rates. Not that high of course as with FLASH.
My great concern is the questionable data provided on the immune response. The TGFb is only a marker, what about Cytokines? There numerous studies suggesting a strong systemic signal. See above. So FLASH damage is it higher as expected? If yes how it is explained the reduced hypothetical adverse effects? Here as the authors discuss there are a mixed set of data on cell killing. If you have increased cell killing then DAMPs are released. That may lead to better tumor control but also increased systemic and abscopal effects. This needs better discussion.
Reviewer 3 Report
The authors have reviewed the current literature available on FLASH radiation. They discuss the potential of the technology for cancer treatment, shortcomings in pre-clinical studies, potential mechanisms underlying the beneficial FLASH effect, and potential for FLASH PBT as the clinical transfer of the technology. There have been a few review papers on FLASH already this year and last year. However, this paper is well written, is up-to-date, and focuses a bit more on the potential of PBT than other review papers. So, there might still be a role to play for this paper.
The authors seem to have been inspired a lot by a previous review paper by Wilson et al. (https://doi.org/10.3389/fonc.2019.01563). At least, this review paper discusses many of the same sub-topics and draws the same conclusions. A reference to this review needs to be added. The authors present only one figure, which is far from original. A similar figure accompanies almost all talks and papers on proton therapy. I can accept that but at least complete the figure by putting scales and units on the axes and energies on the different curves (photon, electron and proton curves).
As evident in the minor comments below, I mainly have issues with the introduction.
Minor comments:
Line 13: Change “…ultra-high dose and dose rate radiation…” to ”… ultra-high dose rate radiation…”.
Line 20: “…have utilised low energy electron beams…”. The electron beams used clinically (4-25 MeV) are generally defined as “high energy electron beams”. Hence, remove “low energy”.
Lines 33-35: “This occurs because radiotherapy utilises an external radiation beam which deposits energy exponentially throughout the depth of the patient tissue [4]”. This is only completely true for keV X-ray beams. For MeV beams, you have a build-up part of the dose before it falls of exponentially, see your figure 1. Furthermore, on lines 35-38: “Therefore, in the cases of deep seated tumours, the healthy normal tissue in front of the tumour receives a large dose of ionising radiation relative to the tumour. Furthermore, healthy normal tissue located behind the tumour can receive an exit dose of radiation if the beam passes through the tumour.” This is a poor description of modern radiotherapy that utilises highly conformal VMAT technologies to deliver the dose. I understand that you want to highlight the benefit of proton radiotherapy but this way of doing it is highly biased towards proton therapy. This section needs to be rephrased!
Lines 39-43: “Advancements in modern radiotherapy deliverance and imaging techniques such as dose fractionation, image-guided radiotherapy, intensity-modulated radiotherapy, volumetric modulated arc therapy and multi-leaf collimation, along with targeted combinatorial drug therapies and immunotherapy, have increased the therapeutic index of radiotherapy [5-10].” Dose fractionation is as old as radiotherapy and multi-leaf collimation is at least 30 years old. This part needs to be rephrased!
Line 44: “…can be specifically targeted at the tumour…”. This is not truer for proton therapy than photon therapy. You always specifically target the tumour, or at least the target volume. The benefit with protons is the finite range and somewhat the lower entrance dose, the result of which is a much smaller volume exposed to lower levels of the radiation (the “dose bath”). The high dose conformity is rarely better for proton therapy. Needs to be rephrased!
Line 49: I have never heard the word “spate” before. It is likely not wrong to use it but I would urge you to use a more well-known synonym, e.g. “series of research studies”.
Line 52: Change “…of high dose at ultra-high dose rates…” to “of dose at ultra-high dose rates”.
Line 53-54: Change “…than conventional doses and dose rates (~5 Gy/min, multiple ~2 Gy fractions over several weeks)…” to “than conventional dose rates (~5 Gy/min)…”.
Line 71: Change “…peaked recently…” to “…peaked more recently…”.
Line 98: “…that bear more preclinical relevance…” needs to rephrased, e.g. “…that are more similar to humans”.
Lines 154-156: “The exact biochemical mechanisms that result in the FLASH effect are yet to be fully elucidated, although the current theory gaining the most ground implicates oxygen as a critical molecule in the biological response to FLASH radiation.” A reference here would be good, e.g. to the review paper by Wilson et al. (https://doi.org/10.3389/fonc.2019.01563) as mentioned above.
Line 181: Change “…conventional dose irradiation…” to “…conventional dose rate irradiation…”.
Line 201: Change “…liable iron…” to “…labile iron…”
Line 242: “…accelerators [12,14,15,54].” Add reference to paper by Lempart et al. (https://doi.org/10.1016/j.radonc.2019.01.031).
Line 244: Change “…of these low energy beams…” to “…of these beams…”.
Lines 244-245: “…PBT offers a much greater tissue penetration…” This will of course depend on the beam energy. Clarify this, e.g. by saying “clinical PBT” or “clinically used PBT” or something along those lines.
Lines 245-247: “The advantages of PBT over conventional photon radiotherapy are already widely recognised in the oncology field.” This is not true! Proton treatment facilities struggle as very little evidence exist showing the benefit of proton therapy over modern photon therapy. Because of this, insurance companies refuse to pay the extra cost that goes with proton treatment compared to photon treatment, the exception being treatment of children. I would rephrase or remove this statement!
Lines 248-250: “…allowing the precise targeting of the tumour volume whilst sparing normal healthy tissue and organs, which contrasts with photons and electrons [11].” This is a highly simplified way of comparing treatment with protons to that of electrons and photons. This is more of the basis on which the treatment planning is carried out. You should limit your statements regarding this figure to depth-dose distributions, which is of course something that needs to be considered when planning a treatment. The advantage of proton treatment to that of photon treatment diminishes as soon as you move away from this one-beam situation, which is hardly a relevant way of comparing modern treatment techniques/sources.
Figure 1: As previously mentioned, add scales and units to the axes and add the energy of the different beams displayed. X-axis title should be “Depth in tissue”.
